# Leverage point themes within Dutch municipalities' healthy weight approaches: A qualitative study from a systems perspective

Maud J. J. ter Bogt[1,2]*, Kirsten E. Bevelander[1,2‡], Lisa Tholen[1], Gerard R. M. Molleman[1,2‡], Maria van den Muijsenbergh[1,3‡], Gerdine A. J. Fransen[1,2]

**1** Primary and Community Care, Radboud University Medical Centre, Nijmegen, The Netherlands, **2** AMPHI Academic Collaborative Centre, Nijmegen, The Netherlands, **3** Pharos, The Dutch Centre of Expertise on Health Disparities, Utrecht, The Netherlands

☯ These authors contributed equally to this work.
‡ KEB, GRMM and MM also contributed equally to this work.
* maud.terbogt@radboudumc.nl.

**Data Availability Statement:** The participant group is so small (n=5) that reidentification is a real risk and would render anonymization of the qualitative

## Abstract

### Introduction

Despite all efforts of national and local approaches, obesity rates continue to rise worldwide. It is increasingly recognized that the complexity of obesity should be further addressed by incorporating a systems perspective when implementing approaches. Such an approach has four interconnected system levels: events, structures, goals, and beliefs, in which small changes ('leverage points') can lead to substantial changes in the functioning of the entire system. The current research examined the functioning of five Dutch municipalities' healthy weight approaches (HWAs) and the leverage point themes that can be identified in their system.

### Methods

Thirty-four semi-structured interviews were conducted with various stakeholders about the HWA, including policy advisors, care professionals, practice professionals, and citizens. An inductive thematic analysis was performed.

### Results

Three main themes were identified: 1) HWA organization structure, 2) collaboration between professionals, and 3) citizen participation. Across all system levels, we identified leverage point themes. The upper-levels events and structures occurred the most and were explained by underlying goals and beliefs. Leverage point themes regarding "HWA organization structure" were municipal processes, such as perceived impact; diversity of themes, activities, and tasks; network; and communication strategies, such as messages about the HWA. Leverage point themes regarding "collaboration between professionals" were linking pins, indicating central players within the network; motivation and commitment including support base; and stimulating one another to work on the HWA by spurring other professionals into

data impossible. Our data consists of interview transcripts with traceable information, because we have spoken to stakeholders of five municipalities. Even if we would leave out the participant's function name or the name of the municipality, the content does not guarantee anonymity due to the specific details that are spoken about. For example, the activity names or other characteristics of a HWA can be traced back to certain municipalities. Consequently, neither the raw transcripts nor the extracts can be shared publicly, even on request. Thus, a persistent identifier cannot be provided for the data. For verification of our results, aggregated data could possibly be provided upon request. Therefore, data may be requested by e-mailing the quality team of our department of primary care (kwaliteitsteam.elg@radboudumc.nl). This is in line with our institute's policy, as we have discussed with our data management officer.

**Funding:** This research was funded by ZonMw. The funders had no role in study design, data collection and analysis, decision to publish, or preparation of the manuscript.

**Competing interests:** The authors have declared that no competing interests exist.

action. Lastly, leverage point themes under "citizen participation" included reaching the target group, e.g., look for entry points; and citizens' motivation, including customization.

## Discussion

This paper provides unique insights into HWAs' leverage point themes that can lead to substantial changes in how the entire system functions and makes suggestions about underlying leverage points to help stakeholders improve their HWA. Future research could focus on studying leverage points within leverage point themes.

## Introduction

Despite all efforts of national and local preventive programs, obesity rates continue to rise worldwide [1, 2]. Recently, there has been a shift in approaches to reduce obesity toward seeing healthy weight as the outcome of obesogenic environments rather than purely the responsibility of individuals [3, 4]. An obesogenic environment is defined as "the sum of influences that the surroundings, opportunities, or conditions of life have on promoting obesity in individuals or populations" [5]. In the Netherlands, municipalities are responsible for public health. Many municipalities and municipal health services try to influence the obesogenic environment by implementing policies, preventive programs, and healthy weight approaches [e.g., 6, 7], yet their impact is limited [8, 9]). Therefore, approaches need to be further improved. For example, in the Netherlands various interventions designed to improve food environments among adolescents were implemented in different contexts, but these interventions showed little evidence on their own. These findings suggest the need of combined interventions that together embrace the complexity of healthy weight [8]. Generally, a healthy weight approach (HWA) consists of elements that promote physical activity and a healthy dietary intake and reduce citizens' sedentary behavior. For example, an approach can promote the use of physical living environment facilities (e.g., accessible paths for walking) [10], execute interventions, and stimulate activities to achieve a healthy lifestyle (e.g., combined lifestyle interventions or consultations with a dietician) [11] organized by people that work together in working groups [12]; or it may be woven into regular events (e.g., offering healthy lunches) [13].

HWAs already include integral perspectives, meaning that cooperation is sought with multiple policy areas, sectors, and organizations to work together on diverse health themes from an individual and an environmental perspective [e.g.,14, 15]. However, the literature suggests that HWA effectiveness can be increased by incorporating a systems perspective [4, 16, 17], which constitutes one step further than an integral perspective. A systems perspective sees the obesogenic environment as a complex adaptive system, meaning that the system is non-linear, hard to control, and adaptable over time in unpredictable ways with regard to prevailing social norms, policies, and economic interests [16, 18]. All elements in a complex adaptive system are connected, creating interdependency and feedback [16, 18]. It is increasingly recognized that the complexity of obesity should be addressed further to change the current obesogenic environment into an environment that encourages healthy weight [16, 17, 19–21].

Ideally, stakeholders in a systems approach anticipate the adaptiveness of the system by context-specific actions that complement and influence one another [16]. An example of a program that adopted a systems perspective is the Amsterdam HWA, which aims to improve children's diet, physical activity, and sleep [22]. In this approach, the public, private, voluntary, and community sectors act collectively to address obesity [21]. Furthermore, implemented interventions take account of underlying determinants, interactions, and mechanisms of

obesity in individual, environmental, and behavior change [17, 23]. To achieve this, collaborative relationships, trust between stakeholders, and evaluation in the context of the system, rather than isolated activities, are important [16]–achieved, for example, by incorporating a learning approach in which all stakeholders come together regularly to enable the program's adaptation in response to changes in the complex adaptive system [22]. Adequate leadership, communication, community engagement, good practice examples, monitoring and evaluation systems, organizational development, and accountability are required to facilitate the incorporation of these elements [4, 24].

Within the aforementioned elements of a systems approach, several theoretical models indicate that multiple levels in each system exist [e.g. 25, 26]. Nobles et al. (2021) developed the Action Scales Model, which fits within the HWA context because the model was based on practice [16]. The model consists of four systems levels, which range from superficial to deep, and include (1) events: observable behaviors and outcomes of stakeholders within the system, (2) structures: the organization of the system that causes these events to occur, (3) goals: the ambitions toward which the system works, and (4) beliefs: stakeholders' deeply held norms, attitudes, and values about system elements [16]. The four levels are interconnected, and changes in deeper levels are likely to change superficial levels as well. In any of the four levels, leverage point themes (LPTs) may exist. LPTs are directions toward a desired system change and consist of specific underlying leverage points (LPs). LPs are small changes within LPTs that can lead to substantial changes in how the entire system functions [16, 26]. Studying an HWA from a systems perspective may reveal LPs regarding how the HWA can be improved. These LPs can help stakeholders to develop HWAs further.

In five Dutch municipalities (ranging from 19,000 to 41,000 inhabitants) in Gelderland province in the Netherlands, overweight and obesity rates rose to between 49% and 58% in 2020 [27]. Three municipalities had a higher average overweight percentage among adults compared to the national average of 50% [27, 28]. Therefore, various HWA stakeholders expressed the wish to strengthen the current approach. To enable stakeholders to strengthen the HWA, we first wanted to get insight into the current approach from a systems perspective and identify LPTs. Therefore, the following research question was posed: How do the five Dutch municipalities' HWAs function according to stakeholders based on the four system levels (events, structures, goals and beliefs) and which LPT can be identified in the system?

## Methods

### Study design

Semi-structured interviews with HWA stakeholders in the five municipalities took place between April and May 2022. The research ethics committee METC Oost-Nederland approved this research (2021–13172). All participants received an information letter and gave written consent.

### Study setting

In the National Prevention Agreement (NPA) of 2018, the Dutch government set priorities regarding healthy weight, smoking, and mental wellbeing [29–31]. The government's aim regarding the NPA is to reduce overweight and obesity in the Netherlands from 50% to 38% by 2040. In line with the NPA, 300 municipalities developed a Regional or a Local Prevention Agreement whereby municipalities commit themselves to at least one of the three themes [32]. Consequently, the municipalities can strengthen local cooperation and make further agreements with local partners about efforts to promote prevention and reduce health inequalities in their municipality [29–31].

Learning communities form an environment where exchange of knowledge, inspiration, and creating plans is crucial. In the current study, stakeholders in the five municipalities' HWAs have been participating in a four-year project where they formed learning communities to optimize their HWA. The learning communities followed the observe-reflect-plan-act cyclus in two to four yearly meetings. Observations from the HWA were discussed and reflected upon. Afterwards, professionals created and executed plans.

## Study sample

In 2021, stakeholders who were actively involved in the HWA approach and known by the health brokers of the municipal health service or municipality policy advisors were personally invited to participate in a learning community. They were approached by the health brokers or policy advisors via e-mail, telephone or face-to-face. All professionals and citizens who were part of the learning community received an invitation via e-mail in March 2022 to participate in a semi-structured 1-on-1 interview about the HWA. In addition, the municipal health service's health brokers who were not members of the learning community were invited, given their broad knowledge on, and experiences with, the HWA. See Table 1 for the participants' job descriptions. If potential participants did not respond after two weeks, they received a reminder e-mail. If they did not respond to the reminder e-mail within one week, they were phoned.

## Procedure

The interviews took place in a videocall via MS Teams and lasted approximately 45 minutes (LT, female, trained MSc student). If learning community members changed jobs between January and May 2022, both the new and the old members participated in the same interview together. A semi-structured interview guide was designed based on the four system levels (events, structures, goals, beliefs) of the Action Scales Model [16]. The interview guide was pilot-tested twice among health brokers of municipalities other than the five participating in this study (interview protocol in S1 Appendix). The pre-test resulted in minor changes in the formulation of a few follow-up questions. In addition, the interviewer made notes about the mentioned activities per level, organizations, and points of attention, to enable a thorough interview.

First, the HWA system was explained to participants as everything that takes place in a municipality that promotes or hinders a healthy weight among citizens. Furthermore, participants were asked to give a short description of their function. Second, participants were asked about the activities (events) in which they were involved, and how. Third, questions about structures were asked, per event, about the organizations and functions with which the participants worked, why they worked together, and how this cooperation was experienced. Fourth, related to goals, participants were invited to describe their ideal municipality and what they were currently doing to achieve this ideal situation. For every organization or function with which the participants mentioned working, they were asked what their goal was and how important the HWA was to them. Lastly, participants were asked what went well in the HWA and what attention points were identified on any of the system levels. Per attention point, they were encouraged to explain what (new) action could solve it and what organizations should take responsibility for the action. Beliefs were deduced from the abovementioned questions.

## Analysis

Interviews were voice-recorded and transcribed ad verbatim by the researchers and an external transcription agency, and the transcripts were checked for correctness. Transcripts were

**Table 1. Participant description.**

| Function | Organization | Description | n |
|---|---|---|---|
| **Policy advisor** | | | |
| Policy advisor | Municipality | Sets goals about what the municipality aims to achieve; manages and connects local and regional parties related to health; advises the municipal college and board on health policy and the HWA. | 6 (incl. 1 successor) |
| **Health broker** | | | |
| Health broker adults and older people | Municipal health service | Works as project leader and advisor together with other care professionals, organizations, or volunteers in the municipality on various topics to promote health of adults and older people (in consultation with the municipality). | 7 (incl. 2 successors) |
| Health broker youth and school | Municipal health service | Supports and advises schools via the national "Healthy School" approach and/or JOGG approach (Huiberts et al., 2022); works on multiple projects in the neighborhood and at municipal level on broad lifestyle themes (e.g., healthy weight, well-being, smoking, sexuality). | 5 |
| JOGG director and neighborhood sports coach | Gym | Booster, driver, connector, networker, and spider in the web in the JOGG approach and sports (Huiberts et al., 2022). | 1 |
| **Care professional** | | | |
| Dietician/weight consultant | Owner/dietician (primary care or self-employed) | Advises adults and children on nutrition intake and/or coaches people to develop a healthier lifestyle/healthy weight. | 5 |
| Youth nurse | Municipal health service | Tracks the development of children from zero to four years old with their families during consultations; answers parents' questions about parenting; refers children to other professionals when needed. | 2 |
| Physiotherapist | Physiotherapy (primary care) | Physiotherapist and owner. | 1 |
| General practitioner | GP and citizens' initiative | Provides curative care among citizens; contact person for problems at the interface of curative care and well-being. | 1 |
| **Practice professional** | | | |
| Recovery trainer | Physiotherapy (primary care) and gym | Works together with dietitians in the field of weight loss; personal training. | 2 (incl. 1 successor) |
| Lifestyle coach and sports | Physiotherapy (primary care) or self-employed | Coaches citizens independence with overweight and quality of life improvement during consultations; provides the combined lifestyle intervention CooL; affiliated with the citizens' initiative. | 2 |
| Welfare worker (incl. 1 neighborhood sports coach) | Social work | Helps citizens regarding mental health, making people or neighborhoods stronger, expanding their social network and reducing loneliness; coordination of the neighborhood sports coaches. | 2 |
| Gym owner | Gym | Sports clubs owner; does not carry out the projects. | 1 |
| **Citizen** | | | |
| Citizen | Citizens' initiatives | Board member of a citizens' initiative; organizes (structural) activities with citizens to have a healthy and pleasant life. | 3 |

analyzed thematically [33], using Atlast.ti version 9 software. Two coders (MB, LT) applied open coding in line with the research question to one transcript while discussing the codes [34]. Afterwards, both coders individually coded the same four interview transcripts inductively, and differences and similarities were then discussed until consensus was reached, resulting in a conceptual coding structure. The conceptual coding structure was discussed by the two coders and one of the co-authors (KB), resulting in small adjustments to the coding. For example, the code "goes well" was used as an additional code for all elements that were explicitly mentioned as going well. The coding structure was then applied to five other transcripts. When new codes emerged during this process, these were discussed with the research team (MB, LT, KB). Next, identical codes were merged by both coders (MB, LT). This resulted in a final coding structure that was consistently applied by one coder until all interviews were coded (LT).

In the next step, the coded data were discussed by the research team (MB, LT, KB), and codes were grouped into subthemes. For example, codes describing each of the health themes

belonging to the HWA (e.g., "exercise more", "broader than weight", "mental", "less over-weight", and "meaningfulness in life") were grouped in the subtheme "diversity of themes". Next, the identified subthemes were discussed, ordered, and categorized under main themes (MB, LT, KB). For example, "municipal processes", "diversity of themes, activities, and tasks", "network", and "communication strategies" all related to the "organization of the HWA". The grouping resulted in three main themes: 1) HWA organization structure, 2) collaboration between professionals, and 3) citizen participation. Lastly, using the written results section, the main author (MB) identified the main LPTs within the subthemes by answering the question "What needs to be changed in this subtheme to strengthen the HWA tremendously?". Besides, the main author (MB) linked every LPT to the main corresponding system levels based on the definitions of events, structures, goals, or beliefs [16]; these were afterwards discussed with one of the supervisors (KB) until consensus was reached. Although the LPTs and system levels are intertwined, only the system levels that were most prevalent in an LPT were included. The results should thus be interpreted from this perspective.

# Results

## Participants

Thirty-four interviews (participation rate 100%) were conducted across five municipalities. Four of the interviews were conducted jointly with the previous employee and the successor (see Table 1).

## Themes

Fig 1 presents an overview of the three main themes, subthemes, and underlying LPTs relating to the four system levels (illustrated by the square brackets for LPTs). In this results section, the subthemes and underlying LPTs are described. For example, within the main theme "HWA organization structure", the subtheme "municipal processes" consists of the LPTs "prioritize health", "slow process", "unclear HWA", "financial resources", and "perceived impact".

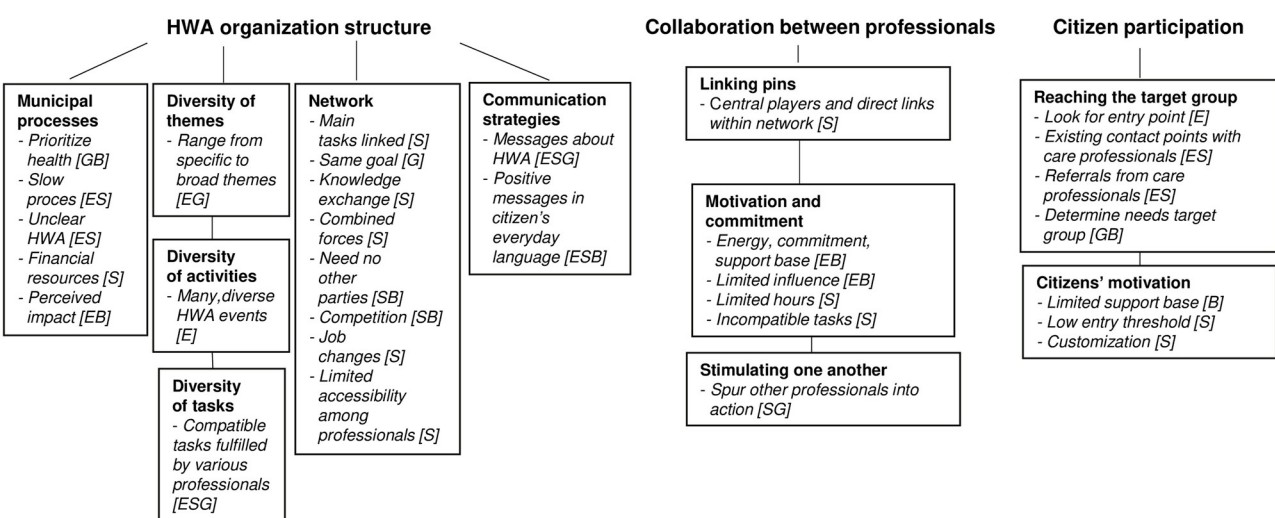

**Fig 1. Overview of main themes, subthemes (bold), underlying LPTs (in bullets), and links to the Action Scales Model (between square brackets).**
*Note*: *HWA = healthy weight approach; E = events; S = structures; G = goals; B = beliefs* [16].

## HWA organization structure

Within the main theme "HWA organization structure", the following subthemes were identified: municipal processes; diversity of themes, activities, and tasks; network; and communication strategies.

**Municipal processes.** Municipal processes concerned mainly municipalities' and stakeholders' roles regarding the formulation and execution of HWA policy. Professionals indicated that, when the municipalities prioritized health (LPT—prioritize health), this facilitated the formulation of policy objectives. Subsequently, some municipalities had elaborated policy documents about healthy weight, whereas others had not. The municipality developed policy objectives on healthy weight, sometimes together with other stakeholders. The policy objectives helped policy advisors to provide direction and guidance in line with the municipality's vision. The municipality acted as contracting authority and was the principal of the activities resulting from the policy objectives, and stakeholders such as health brokers acted as agents of the principal and put these policy objectives in practice.

> *Because it [the working group] belongs to the prevention agreement, as a municipality I am actually the principal for the prevention agreement. [Name health broker] is actually the chairman, the leader of that club. In the working group, I am on the one hand the principal of the health broker adults, but on the other hand I am also just a co-thinker.*
>
> —Policy advisor

Stakeholders also experienced barriers related to municipal processes, as the processes were perceived as slow (LPT—slow process) and professionals sometimes found the HWA unclear (LPT—unclear HWA). For example, professionals indicated that municipalities' topics and visions were perceived as too broad. Another barrier was too few financial resources (LPT—financial resources), which affected the number of working hours and the sustainability of events. The HWA's impact was perceived as limited (LPT—perceived impact), as a result of limited structural funding and the difficulty of expressing the HWA's impact in terms of citizens' health gains. Still, monitoring, evaluation, and reflection activities occurred on different levels (e.g., municipal, school, or activity level) but were sometimes forgotten by practice or care professionals. Some professionals wanted more municipality' actions to ensure continuation by arranging finances and including the activity in policies. For example, the municipalities' sport and prevention agreements were a source of funding for the HWA, but these agreements lasted only a few years:

> *For example, [name intervention], that's every year the question of whether there is money somewhere, until now we succeeded every year. I tend to think that it will always be like this, but that is not necessary at all.*
>
> —Health broker

**Diversity of themes, activities, and tasks.** Stakeholders often focused on a specific theme and target group. Health brokers focused primarily on themes and activities in line with the policy goals, whereas practice professionals focused mainly on their personal task and function goals. For example, a dietician focused mainly on nutrition, whereas a social worker focused mainly on mental health, independent of the policy goals. Therefore, HWA themes ranged from specific (e.g., more exercise and healthier eating) to broad (e.g., broad development among children, meaningfulness in life) (LPT—range from specific to broad themes).

Across this diversity of themes, participants found that many and diverse initiatives and activities (LPT—many, diverse HWA events) were organized, thereby facilitating the integral aspects of the HWA. For example, the HWA consisted of activities (e.g., interventions, handing out flyers), facilities (e.g., cycle paths), working groups (e.g., working groups from the prevention agreement), daily work activities (e.g., information provision, healthy food offers), and other HWA elements (e.g., subsidies). Activities were organized by the municipal health services or local initiatives. However, some participants perceived few initiatives. For example, some villages had no or only small sports associations:

> *[Name municipality] is quite vast, so you don't have in every village, just like in a larger city you have judo and tennis and mountain biking, you name it. Most villages are quite small. And then you only have a football club. In [name municipality] itself, you have something more.*

> —Health broker

Many different organizational tasks in the HWA were fulfilled by different professionals because their regular tasks differed, resulting in a diverse and compatible HWA execution (LPT—compatible tasks fulfilled by various professionals) and facilitating the organization of diverse initiatives. Several participants mentioned that their task depended on other professionals' requests. Health brokers' tasks included organizing HWA activities, thinking along with other professionals based on their own expertise (e.g., in a sounding board group), or supporting citizens and participants with the execution of an activity (e.g., by supporting in preparatory work). Mainly practice and care professionals indicated coaching the target group and helping citizens as soon as the target group was reached. For example, professionals tried to raise awareness among citizens about a healthy lifestyle through presentations, handing out flyers, or consultations.

**Network.**   Regarding HWA organization structure, a relatively small network worked on the HWA within the municipalities. Therefore, professionals often met the same professionals. Participants described having intensive contact with other professionals who had a main task that linked directly to theirs (LPT—main tasks linked). For example, the health brokers had intensive contact with policy advisors, the youth nurse with dieticians, and different health care professionals that worked together within one intervention with one another. Some participants had structural contact in a small fixed group, such as health brokers and the policy advisor in one municipality. The extent to which contact between professionals was perceived as intense determined how and whether professionals executed actions within the HWA (together). Contact between professionals was facilitated by professionals working on the same goal (LPT—same goal), exchanging knowledge (LPT—knowledge exchange), or combining forces to complement one another (LPT—combined forces).

Nevertheless, some practice professionals worked, or found that other professionals worked, independently on the HWA. These professionals indicated that they were not able to find all HWA stakeholders in their municipality and barely collaborated with professionals. Barriers to collaboration included participants feeling that no other parties were needed (LPT—need no other parties), experiencing competition, or believing that a certain target group was"theirs" to work with (LPT—competition). Other collaboration barriers included frequent job changes (LPT—job changes) and limited accessibility among other professionals (e.g., barely responding to questions, feeling that the professional was busy, having to visit one another actively) (LPT—limited accessibility among professionals). These participants were thus not very aware of other HWA activities and HWA professionals:

*I see only my own projects. I don't hear much about new projects or anything. (. . .) Then it all looks a bit like its own island.*

—Practice professional

**Communication strategies.** Communication constituted an important structure within the HWA organization structure (LPT—messages about HWA) to keep the network informed. Therefore, messages about HWA activities were communicated to citizens and professionals via various communication channels, for example local television, newspapers, the internet, or care professionals. Still, participants desired more ways to communicate in order to create more familiarity about activities within the municipalities among both citizens and professionals. Participants stated that communication strategies should include positive messages in citizens' everyday language, not formulated in terms of weight and not patronizing, and spread at places frequented by citizens (LPT—positive messages in citizen's everyday language). Furthermore, participants indicated that all professionals and organizations should take action to convey messages about their part of the HWA themselves and present themselves together with their cooperation partners to create familiarity and show importance and cooperation among others:

*In any case, I think it is very good for schools to present you together, and to show that you work together. (. . .), instead of presenting yourself as individual figures at school (. . .).*

—Health broker

## Collaboration between professionals

The main theme "collaboration between professionals" had three subthemes: linking pins, motivation and commitment, and stimulating one another to work on the HWA.

**Linking pins.** As HWA professionals worked in a network on a variety of tasks and themes with limited available time, more connections between professionals were desired. Therefore, 'linking pins' or 'central players' facilitated collaboration between professionals (LPT—central players and direct links within network). Often these linking pins were health brokers and policy advisors, as they had short communication lines with other participants. Health brokers were the direct link between the organizations and practice professionals on the one hand, and policy advisors on the other hand. Practice professionals were also perceived as central players in the network:

*In principle, I do not need to know all the associations, but if I would walk to the neighborhood sports coach with this question "this gentleman wants to stop doing fitness and wants do to something else, what can I recommend him?" And then I could actually give contact details, from here you have him, he is done with us, further with you, make sure he ends up somewhere where he can do something else fun.*

—Practice professional

**Motivation and commitment.** Once the professionals were linked to one another, the quality of their collaboration depended on their motivation and ability to commit to their HWA tasks. Many stakeholders were enthusiastic about working on specific HWA tasks and experienced a lot of energy, commitment, and support among other professionals (LPT— energy, commitment, support base); this was regarded as a facilitator:

*I find it really amazing how much energy and power can be in such a [working group] meeting and how much enthusiasm. (. . .) And you just notice that selflessly, yes, people are going to do that, hé. So yes, I think that's really special.*

—Care professional

Participants indicated that professionals' level of motivation varied and that many professionals had a core business that was not directly linked to the healthy weight approach:

*My experience is that it varies a lot. That it really depends on the location director of a school or childcare. (. . .) That is very fragmented. (. . .) If you just have a few local ambassadors within a sports association, who are on the board, or volunteers or trainers, that is also the case at schools, that helps enormously.*

—Practice professional

*My experience is that you have to go to those parties yourself. Because for those parties it is not their core business, health, lifestyle, sport, you name it, vitality. Everyone is very busy.*

—Practice professional

Professionals perceived the limited influence on the HWA (LPT—limited influence) that resulted from their dependence on other parties as a barrier to their motivation. For example, at municipal level, the municipal college and council changed every four years, and, at national level, rules were set that limited healthy lifestyles among citizens. Moreover, capacity issues decreased the professionals' level of commitment and execution of HWA tasks. Capacity issues were caused by limited hours (LPT—limited hours) and incompatible tasks (LPT- incompatible tasks); this was often the case when a professional perceived that an HWA task was not his/her priority or (core) task—for example, when professionals focused on their internal organization, such as when sports school employees focused on recruiting members. Furthermore, some participants worked on activities in their spare time, whereas others were paid for such work:

*I am self-employed, so that I often sit there in my spare time. But the municipality has the working hours. (. . ..) I believe I have signed three agreements in the past five years, and every time I have to come to such an evening, then I think yes. There is never any compensation for it.*

—Care professional

**Stimulating one another to work on the HWA.** Some participants spurred other professionals into action to work on the HWA (LPT—spur other professionals into action) by coordinating activities, taking the lead in an action, or chairing a working group. Moreover, practice and care professionals made citizens feel responsible for their own health in order to motivate citizens themselves to act. Still, spurring others into action was perceived as hard:

*Then I see myself as a kind of temporary driving force, because then I hope that at some point it will be taken over by others in the local area. That at some point [I] can focus on other problems or other themes. But that's not so self-evident.*

—Health broker

## Citizen participation

The main theme "citizen participation" had two subthemes: reaching the target group and citizens' motivation.

**Reaching the target group.** Participants tried actively to reach the target group by involving citizens. Therefore, they looked for an entry point into the group (LPT—look for entry point), for example via key persons (e.g., care professionals) or locations (e.g., community centers, sports associations, or schools). Moreover, the fact that existing contact points with care professionals (LPT—existing contact points with care professionals) could be used to talk with citizens about lifestyle or HWA activities facilitated reaching the target group:

*We talk about it [during a consultation], but also about the exercise offer of course. And that children can try out all those sports, that's what we're talking about, of course. About the subsidies that parents can claim if they have a low income, if money is a barrier. And I think they also know that they can go to the park and that they can go to the playground.*

—Care professional

Even though care professionals sometimes referred citizens with overweight to other professionals, participants that carried out activities desired more referrals (LPT—referrals from care professionals):

*We notice that referrals from general practitioners are really difficult to get going, because they are simply extremely busy, they say.*

—Practice professional

Despite their efforts, participants perceived that reaching the target group was hard, but various participants, such as health brokers and practice professionals who were in contact with citizens, stressed the importance of determining citizens' needs in order to adjust the HWA accordingly (LPT—determine needs target group). At the same time, some participants raised questions about who constituted their target group, so they questioned whether focusing on vulnerable citizens or citizens who want to work on lifestyle should be the first step:

*I have that feeling myself, that the people where the most profit can be made are actually the most difficult to reach. Do you want to continue working on that or do you start working on the people who do want to, and try to achieve a kind of snowball effect.*

—Policy advisor

**Citizens' motivation.** Participants found limited support among citizens regarding healthy weight and the HWA (LPT—limited support base), which means that improving citizens' weight status has no priority among citizens and they show limited support for the HWA. Various reasons were mentioned. For example, a lifestyle coach mentioned that citizens did not want to participate in their own municipality because of shame caused by experienced stigma among citizens who have overweight or citizens who go to a dietician. In one municipality, a low support base reflected religious beliefs and culture, as professionals explained that citizens belonging to a particular church were not allowed to engage in competitive sports, but there were few non-competitive sports in the municipality.

Other barriers, perceived by professionals, to citizens taking action on overweight included lack of funds, no meaningfulness, limited self-confidence, and no feeling of urgency.

Therefore, some participants felt that citizens needed a push and indicated that low entry thresholds (LPT—low entry threshold) encouraged citizens to participate in an activity and/or to engage in a healthy lifestyle. For example, a smaller participation fee may lower citizens' entry threshold; in contrast, some participants indicated that, when citizens had to pay some money themselves, this created a sense of personal responsibility. Furthermore, participants indicated that small steps over time were needed and that therefore long-term time investment was needed. Participants also perceived a customized approach (LPT—customization), tailor-made to a citizen's personal situation and preferences, as a facilitator. For example, a practice professional mentioned that the detailed description of combined lifestyle interventions resulted in a mismatch and therefore limited citizen participation:

> *And perhaps offer more room for customization with such a GLI [combined lifestyle intervention]. It is now also quite black and white, the GLI is like this, this is the content, on those different facets it has been proven effectively or assessed. I regularly hear that they [citizens] don't start because it doesn't suit them.*
>
> —Practice professional

The identified system levels of the LPTs may help to understand the LPTs from the participants' perspectives. Within all three main themes, all four system levels were present. Fig 1 shows that all subthemes related to events and/or structure, and most additionally linked to goals and/or beliefs. For example, the subtheme "Stimulating one another to work on the HWA" was linked to structures and goals because it clearly related to organizational structure, and some professionals articulated the goal of spurring others into action to organize the HWA.

## Discussion

By conducting interviews along the Action Scales Model [16], the current research examined the HWAs of five Dutch municipalities in terms of four system levels (events, structures, goals, and beliefs) and identified LPTs to improve the system. Three main themes were identified: 1) HWA organization structure, 2) collaboration between professionals, and 3) citizen participation. Within these themes, HWA barriers related for instance to insufficient financial resources, perceived limited impact, and limited citizens' motivation, and HWA facilitators included a high diversity of themes, activities, and tasks, linking pins within the network, stimulating one another to work on the HWA, and reaching the target group by looking for an entry point and referrals from care professionals. These barriers and facilitators were linked to events, structures, goals, and/or beliefs and represented the interdependent LPTs. Even though LPTs may help to implement systems thinking to a higher degree within HWAs and improve these HWAs, suggestions of LPs in terms of small changes toward a desired LPT may indicate how to get there. Therefore, HWA stakeholders should be aware of these LPTs and implement LPs related to the LPTs within their sphere of influence.

At municipal level, professionals desired clearer steering toward actions, as municipal HWA processes were perceived as slow and unclear. Therefore, the steering process could be optimized. A small change to achieve this may be to communicate clear municipal visions and goals (LP). Previous research also addressed the importance of steering within community-based obesity prevention interventions—often operationalized through steering committees consisting of local leaders and key stakeholders who implement the intervention [35].

At professional level, the organization of the HWA should be perceived as compatible with stakeholders' regular work. Even though professionals indicated that many, diverse HWA

activities and tasks were executed by various professionals, execution of HWA tasks was limited when HWA tasks were perceived to conflict with a professional's personal (core) task. This is in line with results of a previous study in New York that focused on a community-based intervention and also indicated that professionals felt that their "regular" job offered insufficient time to do community work, that they did not feel personally empowered to work on system change issues, and that the work required long-term efforts that exceeded grant periods [36]. Other studies related to HWAs or the obesogenic environment also indicated challenges related to a lack of people, capacity, or hours and a lack of financial resources [37–39]. Therefore, a small change to achieve this could be to align job and HWA task descriptions (LP). Moreover, sufficient capacity for the HWA, such as paid hours, should be created (LP). Besides these material facilitators, stakeholders' motivation and commitment to work on HWA tasks were facilitated by a positive working climate and a support base within the network. Previous research also indicates the importance of a positive working climate. For example, a systematic literature review found that structural and physical empowerment enhances nurses' job satisfaction [40]. It is thus desirable to stimulate a positive working climate in the HWA (LP).

Professionals that stimulated others to work on the HWA and linking pins were crucial to initiate execution of the HWA tasks; systems theories also suggest this, indicating that all system elements should be connected [16, 18, 41]. Therefore, a well-connected network is desirable where professionals work together in an accessible way on, for instance, shared goals and knowledge exchange. Previous social network research also has indicated that connectivity between professionals from various domains is desirable, as a social network is a way through which exchange of, for instance, knowledge takes place [e.g., 42, 43]. Moreover, social network theories refer to linking pins as actors with a high level of centrality, indicating that one professional has the most or closest relationships with other professionals [43]. It is thus recommended that every HWA has stakeholders that function as linking pins who stimulate others to work on the HWA (LP).

At citizen level, professionals found that citizens were hard to reach and had a limited support base regarding healthy lifestyle. Other studies on the effectiveness of HWAs in the Netherlands also indicated that the target group, especially citizens with a low socioeconomic status and a migrant background, were hard to reach [38, 44–46]. Therefore, a mentioned LP included looking for an entry point for the target group and involving them, for example via key figures, care participants, community centers, sports associations, and schools; as also suggested in previous research [46]. However, multiple previous studies have indicated limited skills among professionals to approach the target group, for example because choosing the right words is perceived difficult [44]. Furthermore, professionals perceived that several elements facilitated citizens' participation in a healthy lifestyle or activity, such as low entry thresholds for citizens, customization toward citizens, and professionals' long-term time investment; as also suggested in previous research [46]. Previous research has also indicated that low entry thresholds and customization in terms of program components, practicalities, demands, benefits such as incentives, physician recommendations/referrals, and support are important [47, 48]. Therefore, these elements should be incorporated in HWAs and their activities (LP).

## Strengths and limitations

This study's findings should be interpreted in light of a few limitations. To discover all the experiences of the system, we included a large number of participants from various backgrounds, and interviews focused on HWA topics (events, structures, goals, beliefs) that the

professionals perceived to be important. As the four systems levels within our interview protocol were used regarding the broad HWA rather than specific problems, the current interviews could have been more extensive, yet it is unlikely that more extensive interviews would have resulted in different conclusions. Moreover, applying the Action Scales Model regarding the broad HWA made it harder to make connections between subthemes and system levels. Therefore, the LPTs were linked to the system levels that seemed to be strongest, as this indicates where system change is desired and expected, yet it is debatable whether this is appropriate, as the LPTs and system levels are intertwined. For example, the LPT "diversity of themes" referred mainly to events and goals in this study, as diversity of themes can be seen as observable outcomes of stakeholders (events), and participants described themes related to their policy or function-specific goals (goals). Moreover, "diversity of themes" could potentially also be linked to beliefs, as multiple participants believed that the themes on which they were working were essential within the HWA. Furthermore, the organizational structure apparently facilitated a diversity of themes, which might be linked to structure. It is thus arguable that LPTs should be linked to all system levels. To help stakeholders implement these LPTs and improve HWAs, future research could focus on studying LPs within the LPTs, even though some suggestions have been given in this paper.

Second, when participants described the HWA, new professionals with whom they cooperated were mentioned, but these professionals were not additionally interviewed. Future research could include these professionals as well, possibly resulting in new insights, as these professionals were relatively distanced from the HWAs (e.g., school directors) compared with the participants. Third, stakeholders' perceptions of the HWA system were examined. Therefore, it is uncertain whether the identified LPTs are the actual LPTs in the system, but no other conclusions would have been expected even if this research was not based solely on these perceptions. Lastly, the HWA system was studied in five municipalities in the same region of the Netherlands. It is possible that the system has been experienced differently in other municipalities or countries.

Altogether, the results suggest that, in line with the concept of complex adaptive systems [18], adaptiveness toward the context of the municipality, professionals' own work, and citizens' needs is important. This suggests, according to professionals, that an HWA that is adaptable over time in unpredictable ways is needed. A system science perspective, such as by using the Action Scales Model [16], may help to identify next steps toward system change. Currently, system science in the context of the obesogenic environment tends to focus on citizens' behavior change. However, the obesogenic environment is largely influenced by the behavior of professionals such as policymakers, health brokers, care professionals, and practice professionals. The system surrounding professionals' behavior change is thus also important. To our knowledge, the current study was the first to map LPTs and LPs from an integral professional perspective. The abovementioned organizational LPTs and LPs should be included in HWAs and national and local prevention agreements, as these could help to improve current approaches to system change. Consequently, the system change can contribute to improved cooperation between HWA stakeholders, and strengthened HWAs that enable healthy weight among citizens. On individual level, stakeholders could implement the LPs and LPTs 1) within their sphere of influence and/or current tasks, as this increases the likelihood of successful implementation; and 2) LPs and LPTs that relate to deeper system levels (goals and beliefs), as these are the likely to influence more superficial levels as well and are thus more likely to have impact (Fig 1). For example, municipality policy advisors mainly have sphere of influence related to the subtheme municipal processes. As within this subtheme the LPT "prioritize health" relates to goals and beliefs, municipality policy advisors ideally prioritize this LPT and implement related LPs. Still, LPs regarding events and structures may function as boundary conditions

and are thus also relevant. Also, the abovementioned LPTs and LPs reflect mainly solutions at the upper system levels (events and structure), and this may have presented solutions for deeper levels (goals and beliefs). For example, professionals believed that they had to do their job to earn money but also believed that it was important to contribute to the HWA. As these two beliefs contradicted each other in practice, professionals proposed a solution related to structure: professionals' HWA tasks should align with their personal (core) task. Even though interviewing along the Action Scales Model with a broad variety of HWA stakeholders was an adequate method to portray the system and study LPTs, future research could first present the LPTs to the HWA stakeholders, ideally both professionals and citizens, and then invite them to reflect and plan joint actions (LPs). The LPs chosen and how they are put into practice to create system changes could subsequently be monitored. Moreover, as all LPTs are interdependent, an in-depth system science study is recommended to yield insights into underlying mechanisms and possible unintended consequences, for example via group model-building sessions and causal loop diagrams.

## Supporting information

**S1 Appendix. Interview protocol in English.**
(DOCX)

## Author Contributions

**Conceptualization:** Maud J. J. ter Bogt, Kirsten E. Bevelander, Gerard R. M. Molleman, Maria van den Muijsenbergh, Gerdine A. J. Fransen.

**Data curation:** Maud J. J. ter Bogt, Lisa Tholen.

**Formal analysis:** Maud J. J. ter Bogt, Kirsten E. Bevelander, Lisa Tholen.

**Funding acquisition:** Maud J. J. ter Bogt, Gerdine A. J. Fransen.

**Investigation:** Maud J. J. ter Bogt.

**Methodology:** Maud J. J. ter Bogt, Kirsten E. Bevelander, Gerdine A. J. Fransen.

**Project administration:** Maud J. J. ter Bogt, Gerdine A. J. Fransen.

**Supervision:** Maud J. J. ter Bogt, Kirsten E. Bevelander, Gerard R. M. Molleman, Maria van den Muijsenbergh, Gerdine A. J. Fransen.

**Validation:** Maud J. J. ter Bogt.

**Visualization:** Maud J. J. ter Bogt.

**Writing – original draft:** Maud J. J. ter Bogt.

**Writing – review & editing:** Maud J. J. ter Bogt, Kirsten E. Bevelander, Gerard R. M. Molleman, Maria van den Muijsenbergh, Gerdine A. J. Fransen.

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
