## [Decision Letter · Decision Letter 0]

20 Mar 2023

PONE-D-23-04701Leverage point themes within Dutch municipalities' healthy weight approaches: A qualitative study from a systems perspectivePLOS ONE

Dear Dr. Bogt,

Thank you for submitting your manuscript to PLOS ONE. After careful consideration, we feel that it has merit but does not fully meet PLOS ONE’s publication criteria as it currently stands. Therefore, we invite you to submit a revised version of the manuscript that addresses the points raised during the review process.

We look forward to receiving your revised manuscript.

Kind regards,

Elizabeth McGill

Academic Editor

PLOS ONE

Journal Requirements:

3. Please ensure that you include a title page within your main document. You should list all authors and all affiliations as per our author instructions and clearly indicate the corresponding author.

Additional Editor Comments:

In addition to the comments from the reviewers, please address the following in your revised version:

- The abstract is somewhat hard to follow with the use of multiple abbreviations and parentheses

- A COREQ checklist should be submitted as supplementary material

- Check that all in-text references appear in the reference list; for example, Bartelink et al.,2021 is cited multiple times in the text but does not appear in the reference list.

Reviewers' comments:

Reviewer's Responses to Questions

**Comments to the Author**

1. Is the manuscript technically sound, and do the data support the conclusions?

Reviewer #1: Yes

Reviewer #2: Yes

2. Has the statistical analysis been performed appropriately and rigorously? 

Reviewer #1: No

Reviewer #2: N/A

3. Have the authors made all data underlying the findings in their manuscript fully available?

Reviewer #1: No

Reviewer #2: No

4. Is the manuscript presented in an intelligible fashion and written in standard English?

Reviewer #1: Yes

Reviewer #2: Yes

5. Review Comments to the Author

Reviewer #1: Thank you for allowing me the opportunity to review your manuscript. I enjoyed reading your work and believe this manuscript has the potential to be impactful and make a significant contribution with some further work. The revisions below are provided to strengthen the manuscript. I'm looking forward to receiving a revised manuscript in due course.

Page three, line 56: Can you explain or provide an example about why the impact of these healthy weight approaches are limited?

Page five, line 151: How did you gain access to your participants? There needs to be information around sampling strategy in this section.

Page five, lines 162-163: Were there any challenges that arose as a result of interviewing both the old and new members in the same interview together? Did they influence each others’ answers or perspectives? I can see some connotations moving forward and potential ramifications of doing so which might not always be positive for the HWA.

Page five, lines 165-166: Were any changes made to the interview guide as a result of the pilot interviews?

Page five, line 183: In the abstract you said inductive thematic analysis was used, but that isn’t reflected in the narrative of the analysis sub-section. Please either amend the abstract to reflect the analysis done, or update the narrative in the analysis sub-section to reflect completing an inductive thematic analysis. With a thematic analysis, I would expect to see the work of Braun and Clarke (2006) or Clarke and Braun (2013) cited here.

Page five, line 183: In addition to the above, it could be problematic to have two researchers individually code transcripts. This needs to be highlighted as a potential limitation of the approach, if not already done so.

Page eight: In your results, it would be helpful to include the sub-themes in brackets or square brackets next to the narrative so there’s a clear line of understanding and identification between the Figures and the results narrative. For example, which sections in the narrative relate to prioritize health under the municipal processes theme? This would really strengthen the narrative in the results.

Page 10, line 340-346: This is a really interesting quote and goes deeper than the narrative provided. For example, working together and sharing the same goal goes over an above “creating familiarity and showing importance and cooperation among others”. It would be worth re-visiting and revising the narrative here to support the quote provided.

Page 10, line 352: Forgive my naivety, but what is meant by “linking pins”? Is this a colloquial phrase?

Page 13, line 447: Further information is required around citizens not wanting to participate because of shame. Furthermore, what do you mean when you refer to “low support base”? Is this social support or other forms of support?

Page 13, line 478: The first sentence suggests that Action Scales Model was used to inform the interviews and I assume subsequent analysis, yet in the abstract it says inductive thematic analysis, and there’s no mention of the ASM in the methods section particularly relating to procedure, interview guide, and analysis. Therefore, did you take a deductive or inductive approach to the study design and analysis? This needs clarity throughout the manuscript.

Page 15, line 535: It would be interesting to know if the citizen’s perspective on that first sentence was congruent with what the professionals stated, particularly as three citizens were interviewed as part of this study.

General discussion point: There’s a lot of points where your findings support previous research, but to make sure your work is impactful and novel, it’s important to highlight the key findings not reported elsewhere in the literature. In addition, it is important to indicate why these findings are important for those working within a HWA and within the different levels. Please amend the discussion throughout to reflect these points.

Page 15, line 558: Again, here the Action Scales Model is mentioned, but it does not come through in the methods, approach, or results.

Reviewer #2: A useful paper contributing to understanding of system approaches to action on obesogenic environments. The method is well described and appears robust. The authors have drawn upon Action Scales Model to inform approach, interview questions and design. There are, however, a few areas where further clarify is required to enhance the paper.

Introduction: I would like to see the authors describe Action Scales Model and leverage points as 'an' approach to thinking about systems, rather than as 'the' way to think of systems. Systems thinking/science is a large and varied field. Each tool and approach provides a perspective on complex systems, shining a light on some parts yet shadow on others.

Findings and Discussion: The authors introduce the concept of Leverage Point Theme (LPT), defined on lines 106-108. I struggle to differentiate between LPT as distinct from sub-themes and leverage points. LPTs may need further definition and description within findings. My reading of the literature, LPTs are not something we would find in Action Scale Models or Meadows 12 points to intervene framework, so I am unclear how they link results to underpinning systems frameworks.

I also think the findings and discussion are missing some "so what" and "what next" focus. All system leverage point frameworks contain ideas that action on beliefs, then goals, provides most opportunity for system change, yet are difficult to act upon. Can the Action Scales Model be used to prioritise any leverage points identified? How should HWA be better supported to change obesogenic environments? At the moment the findings appear as a big list, without prioritisation.

6. PLOS authors have the option to publish the peer review history of their article (what does this mean?). If published, this will include your full peer review and any attached files.

Reviewer #1: No

Reviewer #2: **Yes: **Mat Walton

---

## [Author Response · Author response to Decision Letter 0]

11 May 2023

Dear Elizabeth McGill, 

We would like to thank the editor and reviewers for the evaluation of our manuscript “Leverage point themes within Dutch municipalities' healthy weight approaches: A qualitative study from a systems perspective” and for considering our work for publication pending revisions. We are grateful for the opportunity to revise and resubmit our work in which we have paid attention to the editorial comments and addressed the reviewer’s suggestions and comments. Next, we will respond to each comment point-by-point. Within our responses we refer to line numbers of the manuscript with track changes.

Remark: 1. Please ensure that your manuscript meets PLOS ONE's style requirements, including those for file naming. The PLOS ONE style templates can be found at 

RESPONSE: We have updated the style and file names.

Remark: 2. We note that you have indicated that data from this study are available upon request. PLOS only allows data to be available upon request if there are legal or ethical restrictions on sharing data publicly. For more information on unacceptable data access restrictions, please see http://journals.plos.org/plosone/s/data-availability#loc-unacceptable-data-access-restrictions. 

RESPONSE: Our data consists of interview transcripts with traceable information, because we have spoken to stakeholders of five municipalities. Even if we would leave out the participant’s function name or the name of the municipality, the content does not guarantee anonymity due to the specific details that are spoken about. For example, the activity names or other characteristics of a HWA can be traced back to certain municipalities. Therefore, data may be requested by e-mailing the main author (maud.terbogt@radboudumc.nl). This is in line with our institute’s policy, as we have discussed with our data management officer. 

Remark: 3. Please ensure that you include a title page within your main document. You should list all authors and all affiliations as per our author instructions and clearly indicate the corresponding author.

RESPONSE: We have updated our title page according to the instructions.

Remark: 4. Please include captions for your Supporting Information files at the end of your manuscript, and update any in-text citations to match accordingly. Please see our Supporting Information guidelines for more information: http://journals.plos.org/plosone/s/supporting-information.

RESPONSE: We renamed our supporting information and added it to the end of our manuscript

Remark 5: The abstract is somewhat hard to follow with the use of multiple abbreviations and parentheses.

RESPONSE: We limited the use of abbreviations and parentheses in our abstract.

Remark 6: A COREQ checklist should be submitted as supplementary material

RESPONSE: We submitted a COREQ checklist as supplementary material.

Remark 7: Check that all in-text references appear in the reference list; for example, Bartelink et al.,2021 is cited multiple times in the text but does not appear in the reference list.

RESPONSE: We have checked all references.

Remark 8: Page three, line 56: Can you explain or provide an example about why the impact of these healthy weight approaches are limited?

RESPONSE: Healthy weight approaches are complex. Therefore, healthy weight approaches ideally embrace this complexity by further developing towards systems approaches (as indicated in 82-91). We added an example in the manuscript: in the Netherlands various interventions designed to improve food environments among adolescents were implemented in different contexts, but these interventions showed little evidence on their own. These findings suggest the need of combined interventions that together embrace the complexity of healthy weight [8]. (line 67-71).

Remark 9: Page five, line 151: How did you gain access to your participants? There needs to be information around sampling strategy in this section.

RESPONSE: We added an explanation about how the learning community members were recruited in our method section (Study sample line 172-175): In 2021, stakeholders who were actively involved in the HWA approach and known by according to the health brokers of the municipal health service or municipality policy advisors were personally invited to participate in a learning community. They were approached by the health brokers or policy advisors via e-mail, telephone or face-to-face.

Remark 10: Page five, lines 162-163: Were there any challenges that arose as a result of interviewing both the old and new members in the same interview together? Did they influence each others’ answers or perspectives? I can see some connotations moving forward and potential ramifications of doing so which might not always be positive for the HWA.

RESPONSE: We thank the reviewer for this question. We did not encounter challenges by interviewing both the old and the new members in the same interview together. The old member provided almost all information, because the new member just started with his/her job and did not know all aspects of the healthy weight approach yet. We noticed that this was a good way to introduce the new member to the healthy weight approach and the learning community. We did not think it changed the content of the interview in any way.

Remark 11: Page five, lines 165-166: Were any changes made to the interview guide as a result of the pilot interviews?

RESPONSE: The pilot interviews did not lead to major changes in the interview protocol. We only changed the formulation of few of the follow-up questions We added a sentence to explain this in the manuscript (line 191-194).

Remark 12: Page five, line 183: In the abstract you said inductive thematic analysis was used, but that isn’t reflected in the narrative of the analysis sub-section. Please either amend the abstract to reflect the analysis done, or update the narrative in the analysis sub-section to reflect completing an inductive thematic analysis. With a thematic analysis, I would expect to see the work of Braun and Clarke (2006) or Clarke and Braun (2013) cited here.

RESPONSE: We thank the reviewer for this suggestion. We indeed used an inductive thematic analysis. Subsequently, we added a reference to Braun and Clarke (2006) (line 214).

Remark 13: Page five, line 183: In addition to the above, it could be problematic to have two researchers individually code transcripts. This needs to be highlighted as a potential limitation of the approach, if not already done so.

RESPONSE: In qualitative research, coding by multiple coders is called ‘researcher triangulation’, which is regarded as a type of data control. It increases data reliability and validity because more researchers are involved in data coding and interpretation. The discussions between the two coders (LT and MB) and another member of the research team (KB) lead to consensus about the results. Coding and analysing data with multiple researchers is seen as an effective way to reduce error, bias and/or misinterpretation of qualitative data (e.g., Pope et al., 2000 – doi: 10.1136/bmj.320.7227.114). Researcher triangulation is also an item (item 24) in the COREQ checklist for qualitative data (Tong et al., 2007). 

Remark 14: Page eight: In your results, it would be helpful to include the sub-themes in brackets or square brackets next to the narrative so there’s a clear line of understanding and identification between the Figures and the results narrative. For example, which sections in the narrative relate to prioritize health under the municipal processes theme? This would really strengthen the narrative in the results.

RESPONSE: We thank the reviewer for this suggestion. In our manuscript we put LPT between brackets (LPT) when a leverage point theme was described. We now added the name of the leverage point to enhance clarity. For example,(LPT – prioritize health). The subthemes were already visible as headings.

Remark 15: Page 10, line 340-346: This is a really interesting quote and goes deeper than the narrative provided. For example, working together and sharing the same goal goes over an above “creating familiarity and showing importance and cooperation among others”. It would be worth re-visiting and revising the narrative here to support the quote provided.

RESPONSE: We agree with the reviewer that the narrative goes over and above what we have mentioned. Indeed, the narrative also relates to other LPT, such as “Combined forces”, “Spur other professionals into action” and “same goal”. We discuss those topics under the corresponding subthemes. To avoid confusion within this specific subtheme “Communication strategies”, we shortened the quote. 

Remark: Page 10, line 352: Forgive my naivety, but what is meant by “linking pins”? Is this a colloquial phrase?

RESPONSE: We understand the unclarity. Linking pins are the professionals in the network who literally link to many other professionals. They can also be called central players. For further clarity, we added this term as well (line 391).

Remark 16: Page 13, line 447: Further information is required around citizens not wanting to participate because of shame. Furthermore, what do you mean when you refer to “low support base”? Is this social support or other forms of support?

RESPONSE: We added why shame was experienced (line 491-492). Further, "Low support base" means that improving citizens’ weight status has no priority among citizens and they show limited support for the HWA.. We added this in the manuscript (line 488-489)

Remark 17: Page 13, line 478: The first sentence suggests that Action Scales Model was used to inform the interviews and I assume subsequent analysis, yet in the abstract it says inductive thematic analysis, and there’s no mention of the ASM in the methods section particularly relating to procedure, interview guide, and analysis. Therefore, did you take a deductive or inductive approach to the study design and analysis? This needs clarity throughout the manuscript.

RESPONSE: We thank the reviewer for this question. Inductive coding means that a researcher starts from scratch in interpreting raw textual data to develop concepts and themes. It is used to (further) develop theory as opposed to deductive coding, which is used to test already existing theory (Azungah et al., 2018). The Action Scales Model is a new framework (Nobles et al., 2021) and has rarely/not been applied to analyse HWAs. To our knowledge, the current study was the first to map LPTs and LPs from an integral professional perspective. For this reason, we applied an inductive approach in which we identified clusters and patterns (which is called ‘thematic analysis’).

We used the ASM framework to provide participants with a focus or frame in which they could share how they perceived and experienced the HWA (i.e., our raw data). Therefore, our interview guide included ASM topics that participants could openly talk about: events, structures, goals and beliefs (as explained in line 188-189). The concepts and themes that emerged through inductive coding were then linked to the leverage point themes and the ASM. as explained in line 233-234).

Remark 18: Page 15, line 535: It would be interesting to know if the citizen’s perspective on that first sentence was congruent with what the professionals stated, particularly as three citizens were interviewed as part of this study.

RESPONSE: The experience of the three citizens who were involved in citizens initiatives regarding the healthy weight approach were congruent with the professionals. For instance, one citizen who organizes trainings indicated that the citizens who are most in need are the most difficult to reach. We have chosen to not highlight experiences of specific stakeholder groups, because the focus of our study is not to compare stakeholder groups. Therefore, we did not highlight this within our results. 

Remark 19: General discussion point: There’s a lot of points where your findings support previous research, but to make sure your work is impactful and novel, it’s important to highlight the key findings not reported elsewhere in the literature. In addition, it is important to indicate why these findings are important for those working within a HWA and within the different levels. Please amend the discussion throughout to reflect these points.

RESPONSE: We thank the reviewer for these suggestions. We added line 640-641 to our manuscript to explain that our research is innovative because it is the first to use the Action Scales Model to map leverage point themes of the heathy weight approach from a broad, integral professional perspective. 

In addition, we added that implementing these leverage points is important as it may result in systems change (line 643-655). This system change can contributed to improved cooperation between HWA stakeholders, and strengthened HWAs that enable healthy weight among citizens to a higher degree (line 644-646).

Remark 20: Page 15, line 558: Again, here the Action Scales Model is mentioned, but it does not come through in the methods, approach, or results.

RESPONSE: We would like to refer to our response to remark 17. We adjusted the wording of the sentence in line 603-604 in the manuscript to prevent misunderstanding.

Remark 21: Introduction: I would like to see the authors describe Action Scales Model and leverage points as 'an' approach to thinking about systems, rather than as 'the' way to think of systems. Systems thinking/science is a large and varied field. Each tool and approach provides a perspective on complex systems, shining a light on some parts yet shadow on others.

RESPONSE: We agree with the reviewer that there are many approaches to thinking about systems. We did not intend to suggest in the introduction that this is "the" way to think about systems. Therefore, we changed some wording to prevent this implication (line 113-117).

Remark 22: Findings and Discussion: The authors introduce the concept of Leverage Point Theme (LPT), defined on lines 106-108. I struggle to differentiate between LPT as distinct from sub-themes and leverage points. LPTs may need further definition and description within findings. My reading of the literature, LPTs are not something we would find in Action Scale Models or Meadows 12 points to intervene framework, so I am unclear how they link results to underpinning systems frameworks

RESPONSE: We believe that the Action Scales Model can be used to identify leverage points, as also described by Nobles et al., 2021. Yet, the leverage points are based on perceptions, meaning that perceived leverage point themes may be identified. This thus means that we identified perceived leverage points, as described in our limitations section (line 625-628). We define leverage point themes as small changes towards a systems change (as described by Nobles et al., 2021), implying that leverage points are specific. It is debatable to what extent our identified leverage points are specific and small. We believe that leverage point themes are directions toward a desired system change and consist of specific underlying leverage point (as explained in line 122-125). We use the term leverage point themes within our results section, as we believe that more concrete leverage points exist within our leverage point themes. For instance, based on our results section and previous research, we provided examples of leverage points in our discussion, by (LP). The described subthemes resulted from our analysis process, and describe the subthemes within our three main themes that also resulted from our analysis process (as described in line 212-243). With this research, we provided insights into leverage point themes within these subthemes.

Remark 23: I also think the findings and discussion are missing some "so what" and "what next" focus. All system leverage point frameworks contain ideas that action on beliefs, then goals, provides most opportunity for system change, yet are difficult to act upon. Can the Action Scales Model be used to prioritise any leverage points identified? How should HWA be better supported to change obesogenic environments? At the moment the findings appear as a big list, without prioritisation.

RESPONSE: We thank the reviewer for these suggestions and questions. We agree that the results may appear as a big list. We believe that all leverage point themes are important within a systems approach. Nevertheless, prioritisation on an individual or function level could be made, meaning that different stakeholders should prioritize different leverage point themes. Therefore, we have added this to line 644-650. As the healthy weight approach is a broad integral approach where a lot of different stakeholders are involved in, leverage point themes on many different topics exist. For example, a policy maker has a completely different sphere of influence compared to a dietician. Therefore, we recommend that stakeholders implement the leverage point themes within their sphere of influence. As the reviewer stated, actions that are linked towards beliefs and goals may have more impact. Yet, actions that are implemented thoroughly and are mainly linked to events and structures may be impactful as well. Therefore, we recommend that stakeholders implement leverage point themes within their sphere of influence that are ideally linked to their current tasks, as this increases the chances of successful implementation (e.g. due to time). Moreover, the implemented leverage point themes are ideally linked towards the systems’ levels beliefs and/or goals. We revised the text and provided an example (line 644-653) to make this clearer.

We would like to thank you and the reviewers for their thorough evaluation and helpful additional comments. We are confident that we were able to address all concerns and that the manuscript in its current form should meet the standards required for publication. 

On behalf of all authors,

Yours sincerely,

Maud ter Bogt, MSc

Radboud university medical center

Department of Primary and Community Care, Nijmegen, The Netherlands

Geert Grooteplein Noord 21, 6500 HB Nijmegen.

---

## [Decision Letter · Decision Letter 1]

30 May 2023

Leverage point themes within Dutch municipalities' healthy weight approaches: A qualitative study from a systems perspective

PONE-D-23-04701R1

Dear Dr. Bogt,

We’re pleased to inform you that your manuscript has been judged scientifically suitable for publication and will be formally accepted for publication once it meets all outstanding technical requirements.

Kind regards,

Elizabeth McGill

Academic Editor

PLOS ONE

Additional Editor Comments (optional):

Thank you for addressing the reviewers' and editor comments. I look forward to seeing this work published in due course. 

Reviewers' comments:

Reviewer's Responses to Questions

**Comments to the Author**

1. If the authors have adequately addressed your comments raised in a previous round of review and you feel that this manuscript is now acceptable for publication, you may indicate that here to bypass the “Comments to the Author” section, enter your conflict of interest statement in the “Confidential to Editor” section, and submit your "Accept" recommendation.

Reviewer #1: All comments have been addressed

Reviewer #2: All comments have been addressed

2. Is the manuscript technically sound, and do the data support the conclusions?

Reviewer #1: Yes

Reviewer #2: Yes

3. Has the statistical analysis been performed appropriately and rigorously? 

Reviewer #1: Yes

Reviewer #2: N/A

4. Have the authors made all data underlying the findings in their manuscript fully available?

Reviewer #1: No

Reviewer #2: Yes

5. Is the manuscript presented in an intelligible fashion and written in standard English?

Reviewer #1: Yes

Reviewer #2: Yes

6. Review Comments to the Author

Reviewer #1: Thank you for taking on board our feedback and revising the manuscript. I'm happy to accept the manuscript in its current form. Congratulations, team.

Reviewer #2: I thank the authors for giving careful consideration to reviewer comments. I am satisfied that substantive comments have been addressed. I feel that the manuscript does make a contribution to the area of study, in particular a worked example of using the Action Scale Model in practice.

7. PLOS authors have the option to publish the peer review history of their article (what does this mean?). If published, this will include your full peer review and any attached files.

Reviewer #1: No

Reviewer #2: **Yes: **Mat Walton

---

## [Editor Report · Acceptance letter]

2 Jun 2023

PONE-D-23-04701R1 

Leverage point themes within Dutch municipalities' healthy weight approaches: A qualitative study from a systems perspective 

Dear Dr. Bogt:

I'm pleased to inform you that your manuscript has been deemed suitable for publication in PLOS ONE. Congratulations! Your manuscript is now with our production department. 

Kind regards, 

on behalf of

Dr Elizabeth McGill 

Academic Editor

PLOS ONE